# Seq2Tens: An Efficient Representation of Sequences by Low-Rank Tensor Projections

**Csaba Toth**[*]        **Patric Bonnier**[*]        **Harald Oberhauser**[*]

[*]Mathematical Institute, University of Oxford
{toth, bonnier, oberhauser}@maths.ox.ac.uk

## Abstract

Sequential data such as time series, video, or text can be challenging to analyse as the ordered structure gives rise to complex dependencies. At the heart of this is non-commutativity, in the sense that reordering the elements of a sequence can completely change its meaning. We use a classical mathematical object – the free algebra – to capture this non-commutativity. To address the innate computational complexity of this algebra, we use compositions of low-rank tensor projections. This yields modular and scalable building blocks that give state-of-the-art performance on standard benchmarks such as multivariate time series classification, mortality prediction and generative models for video. Code and benchmarks are publically available at https://github.com/tgcsaba/seq2tens.

## 1 Introduction

A central task of learning is to find representations of the underlying data that efficiently and faithfully capture their structure. In the case of sequential data, one data point consists of a sequence of objects. This is a rich and non-homogeneous class of data and includes classical uni- or multi-variate time series (sequences of scalars or vectors), video (sequences of images), and text (sequences of letters). Particular challenges of sequential data are that each sequence entry can itself be a highly structured object and that data sets typically include sequences of different length which makes naive vectorization troublesome.

**Contribution.** Our main result is a generic method that takes a *static feature map* for a class of objects (e.g. a feature map for vectors, images, or letters) as input and turns this into a feature map for sequences of arbitrary length of such objects (e.g. a feature map for time series, video, or text). We call this feature map for sequences Seq2Tens for reasons that will become clear; among its attractive properties are that it (i) provides a structured, parsimonious description of sequences; generalizing classical methods for strings, (ii) comes with theoretical guarantees such as universality, (iii) can be turned into modular and flexible neural network (NN) layers for sequence data. The key ingredient to our approach is to embed the feature space of the static feature map into a larger linear space that forms an algebra (a vector space equipped with a multiplication). The product in this algebra is then used to "stitch together" the static features of the individual sequence entries in a structured way. The construction that allows to do all this is classical in mathematics, and known as the *free algebra* (over the static feature space).

**Outline.** Section 2 formalizes the main ideas of Seq2Tens and introduces the free algebra $\mathrm{T}(V)$ over a space $V$ as well as the associated product, the so-called *convolution tensor product*. Section 3 shows how low rank (LR) constructions combined with sequence-to-sequence transforms allows one to efficiently use this rich algebraic structure. Section 4 applies the results of Sections 2 and 3 to build modular and scalable NN layers for sequential data. Section 5 demonstrates the flexibility and modularity of this approach on both discriminative and generative benchmarks. Section 6 makes connections with previous work and summarizes this article. In the appendices we provide mathematical background, extensions, and detailed proofs for our theoretical results.

## 2  CAPTURING ORDER BY NON-COMMUTATIVE MULTIPLICATION

We denote the set of sequences of elements in a set $\mathcal{X}$ by

$$\mathrm{Seq}(\mathcal{X}) = \{\mathbf{x} = (\mathbf{x}_i)_{i=1,\ldots,L} : \mathbf{x}_i \in \mathcal{X}, \ L \geq 1\} \tag{1}$$

where $L \geq 1$ is some arbitrary length. Even if $\mathcal{X}$ itself is a linear space, e.g. $\mathcal{X} = \mathbb{R}$, $\mathrm{Seq}(\mathcal{X})$ is never a linear space since there is no natural addition of two sequences of different length.

**Seq2Tens in a nutshell.**   Given any vector space $V$ we may construct the so-called free algebra $\mathrm{T}(V)$ over $V$. We describe the space $\mathrm{T}(V)$ in detail below, but as for now the only thing that is important is that $\mathrm{T}(V)$ is also a vector space that includes $V$, and that it carries a non-commutative product, which is, in a precise sense, "the most general product" on $V$.

The main idea of Seq2Tens is that any "static feature map" for elements in $\mathcal{X}$

$$\phi : \mathcal{X} \to V$$

can be used to construct a new feature map $\Phi : \mathrm{Seq}(\mathcal{X}) \to \mathrm{T}(V)$ for sequences in $\mathcal{X}$ by using the algebraic structure of $\mathrm{T}(V)$: the non-commutative product on $\mathrm{T}(V)$ makes it possible to "stitch together" the individual features $\phi(\mathbf{x}_1), \ldots, \phi(\mathbf{x}_L) \in V \subset \mathrm{T}(V)$ of the sequence $\mathbf{x}$ in the larger space $\mathrm{T}(V)$ by multiplication in $\mathrm{T}(V)$. With this we may define the feature map $\Phi(\mathbf{x})$ for a sequences $\mathbf{x} = (\mathbf{x}_1, \ldots, \mathbf{x}_L) \in \mathrm{Seq}(\mathcal{X})$ as follows

(i) lift the map $\phi : \mathcal{X} \to V$ to a map $\varphi : \mathcal{X} \to \mathrm{T}(V)$,

(ii) map $\mathrm{Seq}(\mathcal{X}) \to \mathrm{Seq}(\mathrm{T}(V))$ by $(\mathbf{x}_1, \ldots, \mathbf{x}_L) \mapsto (\varphi(\mathbf{x}_1), \ldots, \varphi(\mathbf{x}_L))$,

(iii) map $\mathrm{Seq}(\mathrm{T}(V)) \to \mathrm{T}(V)$ by multiplication $(\varphi(\mathbf{x}_1), \ldots, \varphi(\mathbf{x}_L)) \mapsto \varphi(\mathbf{x}_1) \cdots \varphi(\mathbf{x}_L)$.

In a more concise form, we define $\Phi$ as

$$\Phi : \mathrm{Seq}(\mathcal{X}) \to \mathrm{T}(V), \quad \Phi(\mathbf{x}) = \prod_{i=1}^{L} \varphi(\mathbf{x}_i) \tag{2}$$

where $\prod$ denotes multiplication in $\mathrm{T}(V)$. We refer to the resulting map $\Phi$ as the Seq2Tens map, which stands short for ***Sequences-2-Tensors***. Why is this construction a good idea? First note, that step (i) is always possible since $V \subset \mathrm{T}(V)$ and we discuss the simplest such lift before Theorem 2.1 as well as other choices in Appendix B. Further, if $\phi$, respectively $\varphi$, provides a faithful representation of objects in $\mathcal{X}$, then there is no loss of information in step (ii). Finally, since step (iii) uses "the most general product" to multiply $\varphi(\mathbf{x}_1) \cdots \varphi(\mathbf{x}_L)$ one expects that $\Phi(\mathbf{x}) \in \mathrm{T}(V)$ faithfully represents the sequence $\mathbf{x}$ as an element of $\mathrm{T}(V)$.

Indeed in Theorem 2.1 below we show an even stronger statement, namely that if the static feature map $\phi : \mathcal{X} \to V$ contains enough non-linearities so that non-linear functions from $\mathcal{X}$ to $\mathbb{R}$ can be approximated as *linear functions* of the static feature map $\phi$, then the above construction extends this property to functions of sequences. Put differently, *if $\phi$ is a universal feature map for $\mathcal{X}$, then $\Phi$ is a universal feature map for* $\mathrm{Seq}(\mathcal{X})$; that is, any non-linear function $f(\mathbf{x})$ of a sequence $\mathbf{x}$ can be approximated as a linear functional of $\Phi(\mathbf{x})$, $f(\mathbf{x}) \approx \langle \ell, \Phi(\mathbf{x}) \rangle$. We also emphasize that the domain of $\Phi$ is the space $\mathrm{Seq}(\mathcal{X})$ of sequences of *arbitrary* (finite) length. The remainder of this Section gives more details about steps (i),(ii),(iii) for the construction of $\Phi$.

**The free algebra $\mathrm{T}(V)$ over a vector space $V$.**   Let $V$ be a vector space. We denote by $\mathrm{T}(V)$ the set of sequences of tensors indexed by their degree $m$,

$$\mathrm{T}(V) := \{\mathbf{t} = (\mathbf{t}_m)_{m \geq 0} \mid \mathbf{t}_m \in V^{\otimes m}\} \tag{3}$$

where by convention $V^{\otimes 0} = \mathbb{R}$. For example, if $V = \mathbb{R}^d$ and $\mathbf{t} = (\mathbf{t}_m)_{m \geq 0}$ is some element of $\mathrm{T}(\mathbb{R}^d)$, then its degree $m = 1$ component is a $d$-dimensional vector $\mathbf{t}_1$, its degree $m = 2$ component is a $d \times d$ matrix $\mathbf{t}_2$, and its degree $m = 3$ component is a degree 3 tensor $\mathbf{t}_3$. By defining addition and scalar multiplication as

$$\mathbf{s} + \mathbf{t} := (\mathbf{s}_m + \mathbf{t}_m)_{m \geq 0}, \quad c \cdot \mathbf{t} = (c\mathbf{t}_m)_{m \geq 0} \tag{4}$$

the set $\mathrm{T}(V)$ becomes a linear space. By identifying $v \in V$ as the element $(0, v, 0, \ldots, 0) \in \mathrm{T}(V)$ we see that $V$ is a linear subspace of $\mathrm{T}(V)$. Moreover, while $V$ is only a linear space, $\mathrm{T}(V)$ carries a product that turns $\mathrm{T}(V)$ into an algebra. This product is the so-called *tensor convolution product*, and is defined for $\mathbf{s}, \mathbf{t} \in \mathrm{T}(V)$ as

$$\mathbf{s} \cdot \mathbf{t} := \big( \sum_{i=0}^{m} \mathbf{s}_i \otimes \mathbf{t}_{m-i} \big)_{m \geq 0} = \big( 1, \mathbf{s}_1 + \mathbf{t}_1, \mathbf{s}_2 + \mathbf{s}_1 \otimes \mathbf{t}_1 + \mathbf{t}_2, \ldots \big) \in \mathrm{T}(V) \tag{5}$$

where $\otimes$ denotes the usual outer tensor product; e.g. for vectors $u = (u_i), v = (v_i) \in \mathbb{R}^d$ the outer tensor product $u \otimes v$ is the $d \times d$ matrix $(u_i v_j)_{i,j=1,\ldots,d}$. We emphasize that like the outer tensor product $\otimes$, the tensor convolution product $\cdot$ is non-commutative, i.e. $\mathbf{s} \cdot \mathbf{t} \neq \mathbf{t} \cdot \mathbf{s}$. In a mathematically precise sense, $\mathrm{T}(V)$ *is the most general algebra that contains $V$; it is a "free construction"*. Since $\mathrm{T}(V)$ is realized as series of tensors of increasing degree, the *free algebra* $\mathrm{T}(V)$ is also known as the *tensor algebra* in the literature. Appendix A contains background on tensors and further examples.

**Lifting static feature maps.** Step (i) in the construction of $\Phi$ requires turning a given feature map $\phi : \mathcal{X} \to V$ into a map $\varphi : \mathcal{X} \to \mathrm{T}(V)$. Throughout the rest of this article we use the lift

$$\varphi(\mathbf{x}) = (1, \phi(\mathbf{x}), 0, 0 \ldots) \in \mathrm{T}(V). \tag{6}$$

We discuss other choices in Appendix B, but attractive properties of the lift 6 are that (a) the evaluation of $\Phi$ against low rank tensors becomes a simple recursive formula (Proposition 3.3, (b) it is a generalization of sequence sub-pattern matching as used in string kernels (Appendix B.3, (c) despite its simplicity it performs exceedingly well in practice (Section 4).

**Extending to sequences of arbitrary length.** Steps (i) and (ii) in the construction specify how the map $\Phi : \mathcal{X} \to \mathrm{T}(V)$ behaves on sequences of length-1, that is, single observations. Step (iii) amounts to the requirement that for any two sequences $\mathbf{x} = (\mathbf{x}_1, \ldots, \mathbf{x}_K), \mathbf{y} = (\mathbf{y}_1, \ldots, \mathbf{y}_L) \in \mathrm{Seq}(V)$, their concatenation defined as $\mathbf{z} = (\mathbf{x}_1, \ldots, \mathbf{x}_K, \mathbf{y}_1, \ldots, \mathbf{y}_L) \in \mathrm{Seq}(V)$ can be understood in the feature space as (non-commutative) multiplication of their corresponding features

$$\Phi(\mathbf{z}) = \Phi(\mathbf{x}) \cdot \Phi(\mathbf{y}). \tag{7}$$

In other words, we inductively extend the lift $\varphi$ to sequences of arbitrary length by starting from sequences consisting of a single observation, which is given in equation 2. Repeatedly applying the definition of the tensor convolution product in equation 5 leads to the following explicit formula

$$\Phi_m(\mathbf{x}) = \sum_{1 \leq i_1 < \cdots < i_m \leq L} \mathbf{x}_{i_1} \otimes \cdots \otimes \mathbf{x}_{i_m} \in V^{\otimes m}, \quad \Phi(\mathbf{x}) = (\Phi_m(\mathbf{x}))_{m \geq 0}, \tag{8}$$

where $\mathbf{x} = (\mathbf{x}_1, \ldots, \mathbf{x}_L) \in \mathrm{Seq}(V)$ and the summation is over non-contiguous subsequences of $\mathbf{x}$.

**Some intuition: generalized pattern matching.** Our derivation of the feature map $\Phi(\mathbf{x}) = (1, \Phi_1(\mathbf{x}), \Phi_2(\mathbf{x}), \ldots) \in \mathrm{T}(V)$ was guided by general algebraic principles, but equation 8 provides an intuitive interpretation. It shows that for each $m \geq 1$, the entry $\Phi_m(\mathbf{x}) \in V^{\otimes m}$ constructs a summary of a long sequence $\mathbf{x} = (\mathbf{x}_1, \ldots, \mathbf{x}_L) \in \mathrm{Seq}(V)$ based on subsequences $(\mathbf{x}_{i_1}, \ldots, \mathbf{x}_{i_m})$ of $\mathbf{x}$ of length-$m$. It does this by taking the usual outer tensor product $\mathbf{x}_{i_1} \otimes \cdots \otimes \mathbf{x}_{i_m} \in V^{\otimes m}$ and summing over all possible subsequences. This is completely analogous to how string kernels provide a structured description of text by looking at non-contiguous substrings of length-$m$ (indeed, Appendix B.3 makes this rigorous). However, the main difference is that the above construction works for arbitrary sequences and not just sequences of discrete letters. Readers with less mathematical background might simply take this as motivation and regard equation 8 as definition. However, the algebraic background allows to prove that $\Phi$ is universal, see Theorem 2.1 below.

**Universality.** A function $\phi : \mathcal{X} \to V$ is said to be *universal for $\mathcal{X}$* if all continuous functions on $\mathcal{X}$ can be approximated as linear functions on the image of $\phi$. One of the most powerful features of neural nets is their universality (Hornik, 1991). A very attractive property of $\Phi$ is that it preserves universality: if $\phi : \mathcal{X} \to V$ is universal for $\mathcal{X}$, then $\Phi : \mathrm{Seq}(X) \to \mathrm{T}(V)$ is universal for $\mathrm{Seq}(\mathcal{X})$. To make this precise, note that $V^{\otimes m}$ is a linear space and therefore any $\ell = (\ell_0, \ell_1, \ldots, \ell_M, 0, 0, \ldots) \in$

T(V) consisting of M tensors $\ell_m \in V^{\otimes m}$, yields a linear functional on T(V); e.g. if $V = \mathbb{R}^d$ and we identify $\ell_m$ in coordinates as $\ell_m = (\ell_m^{i_1,\ldots,i_m})_{i_1,\ldots,i_m \in \{1,\ldots,d\}}$ then

$$\langle \ell, \mathbf{t} \rangle := \sum_{m=0}^{M} \langle \ell_m, \mathbf{t}_m \rangle = \sum_{m=0}^{M} \sum_{i_1,\ldots,i_m \in \{1,\ldots,d\}} \ell_m^{i_1,\ldots,i_m} \mathbf{t}_m^{i_1,\ldots,i_m}. \tag{9}$$

Thus linear functionals of the feature map $\Phi$, are real-valued functions of sequences. Theorem 2.1 below shows that any continuous function $f : \mathrm{Seq}(\mathcal{X}) \to \mathbb{R}$ can by arbitrary well approximated by a $\ell \in \mathrm{T}(V)$, $f(\mathbf{x}) \approx \langle \ell, \Phi(\mathbf{x}) \rangle$.

**Theorem 2.1.** *Let $\phi : \mathcal{X} \to V$ be a universal map with a lift that satisfies some mild constraints, then the following map is universal:*

$$\Phi : \mathrm{Seq}(\mathcal{X}) \to \mathrm{T}(V), \quad \mathbf{x} \mapsto \Phi(\mathbf{x}). \tag{10}$$

A detailed proof and the precise statement of Theorem 2.1 is given in Appendix B.

## 3    APPROXIMATION BY LOW-RANK LINEAR FUNCTIONALS

**The combinatorial explosion of tensor coordinates and what to do about it.**    The universality of $\Phi$ suggests the following approach to represent a function $f : \mathrm{Seq}(\mathcal{X}) \to \mathbb{R}$ of sequences: First compute $\Phi(\mathbf{x})$ and then optimize over $\ell$ (and possibly also the hyperparameters of $\phi$) such that $f(\mathbf{x}) \approx \langle \ell, \Phi(\mathbf{x}) \rangle = \sum_{m=0}^{M} \langle \ell_m, \Phi_m(\mathbf{x}) \rangle$. Unfortunately, tensors suffer from a combinatorial explosion in complexity in the sense that even just storing $\Phi_m(\mathbf{x}) \in V^{\otimes m} \subset \mathrm{T}(V)$ requires $O(\dim(V)^m)$ real numbers. Below we resolve this computational bottleneck as follows: in Proposition 3.3 we show that for a special class of low-rank elements $\ell \in \mathrm{T}(V)$, the functional $\mathbf{x} \mapsto \langle \ell, \Phi(\mathbf{x}) \rangle$ can be efficiently computed in both time and memory. This is somewhat analogous to a kernel trick since it shows that $\langle \ell, \Phi(\mathbf{x}) \rangle$ can be cheaply computed without explicitly computing the feature map $\Phi(\mathbf{x})$. However, Theorem 2.1 guarantees universality under no restriction on $\ell$, thus restriction to rank-1 functionals limits the class of functions $f(\mathbf{x})$ that can be approximated. Nevertheless, by iterating these "low-rank functional" constructions in the form of sequence-to-sequence transformations this can be ameliorated. We give the details below but to gain intuition, we invite the reader to think of this iteration analogous to stacking layers in a neural network: each layer is a relatively simple non-linearity (e.g. a sigmoid composed with an affine function) but by composing such layers, complicated functions can be efficiently approximated.

**Rank-1 functionals are computationally cheap.**    Degree $m = 2$ tensors are matrices and low-rank (LR) approximations of matrices are widely used in practice (Udell & Townsend, 2019) to address the quadratic complexity. The definition below generalizes the rank of matrices (tensors of degree $m = 2$) to tensors of any degree $m$.

**Definition 3.1.** The *rank* (also called *CP rank* (Carroll & Chang, 1970)) of a degree-$m$ tensor $\mathbf{t}_m \in V^{\otimes m}$ is the smallest number $r \geq 0$ such that one may write

$$\mathbf{t}_m = \sum_{i=0}^{r} \mathbf{v}_i^1 \otimes \cdots \otimes \mathbf{v}_i^m, \quad \mathbf{v}_i^1, \ldots, \mathbf{v}_i^m \in V. \tag{11}$$

We say that $\mathbf{t} = (\mathbf{t}_m)_{m \geq 0} \in \mathrm{T}(V)$ has rank-1 (and degree-$M$) if each $\mathbf{t}_m \in V^{\otimes m}$ is a rank-1 tensor and $\mathbf{t}_i = 0$ for $i > M$.

**Remark 3.2.** For $\mathbf{x} = (\mathbf{x}_1, \ldots, \mathbf{x}_L) \in \mathrm{Seq}(V)$, the rank $r_m \in \mathbb{N}$ of $\Phi_m(\mathbf{x})$ satisfies $r_m \leq \binom{L}{m}$, while the rank and degree $r, d \in \mathbb{N}$ of $\Phi(\mathbf{x})$ satisfy $r \leq \binom{L}{K}$ for $K = \lfloor \frac{L}{2} \rfloor$ and $d \leq L$.

A direct calculation shows that if $\ell$ is of rank-1, then $\langle \ell, \Phi(\mathbf{x}) \rangle$ can be computed very efficiently by inner product evaluations in $V$.

**Proposition 3.3.** *Let $\ell = (\ell_m)_{m \geq 0} \in \mathrm{T}(V)$ be of rank-1 and degree-$M$. If $\phi$ is lifted to $\varphi$ as in equation 6, then*

$$\langle \ell, \Phi(\mathbf{x}) \rangle = \sum_{m=0}^{M} \sum_{1 \leq i_1 < \cdots < i_m \leq L} \prod_{k=1}^{m} \langle \mathbf{v}_k^m, \phi(\mathbf{x}_{i_k}) \rangle \tag{12}$$

*where $\ell_m = \mathbf{v}_1^m \otimes \cdots \otimes \mathbf{v}_m^m \in V^{\otimes m}$, $\mathbf{v}_i^m \in V$ and $m = 0, \ldots, M$.*

Note that the inner sum is taken over all non-contiguous subsequences of $\mathbf{x}$ of length-$m$, analogously to $m$-mers of strings and we make this connection precise in Appendix B.3; the proof of Proposition 3.3 is given in Appendix B.1.1. While equation 12 looks expensive, by casting it into a recursive formulation over time, it can be computed in $O(M^2 \cdot L \cdot d)$ time and $O(M^2 \cdot (L+c))$ memory, where $d$ is the inner product evaluation time on $V$, while $c$ is the memory footprint of a $v \in V$. This can further be reduced to $O(M \cdot L \cdot d)$ time and $O(M \cdot (L+c))$ memory by an efficient parametrization of the rank-1 element $\ell \in \mathrm{T}(V)$. We give further details in Appendices D.2, D.3, D.4.

**Low-rank Seq2Tens maps.** The composition of a linear map $\mathcal{L} : \mathrm{T}(V) \to \mathbb{R}^N$ with $\Phi$ can be computed cheaply in parallel using equation 12 when $\mathcal{L}$ is specified through a collection of $N \in \mathbb{N}$ rank-1 elements $\ell^1, \ldots, \ell^N \in \mathrm{T}(V)$ such that

$$\tilde{\Phi}_{\tilde{\theta}}(\mathbf{x}_1, \ldots, \mathbf{x}_L) := \mathcal{L} \circ \Phi(\mathbf{x}_1, \ldots, \mathbf{x}_L) = (\langle \ell^j, \Phi(\mathbf{x}_1, \ldots, \mathbf{x}_L) \rangle)_{j=1}^N \in \mathbb{R}^N. \tag{13}$$

We call the resulting map $\tilde{\Phi}_{\tilde{\theta}} : \mathrm{Seq}(\mathcal{X}) \to \mathbb{R}^N$ a ***Low-rank Seq2Tens*** map of width-$N$ and order-$M$, where $M \in \mathbb{N}$ is the maximal degree of $\ell^1, \ldots, \ell^N$ such that $\ell_i^j = 0$ for $i > M$. The LS2T map is parametrized by (1) the component vectors $\mathbf{v}_{j,m}^k \in V$ of the rank-1 elements $\ell_m^j = \mathbf{v}_{j,m}^1 \otimes \cdots \otimes \mathbf{v}_{j,m}^m$, (2) by any parameters $\theta$ that the static feature map $\phi_\theta : \mathcal{X} \to V$ may depend on. We jointly denote these parameters by $\tilde{\theta} = (\theta, \ell^1, \ldots, \ell^N)$. In addition, by the subsequent composition of $\tilde{\Phi}_{\tilde{\theta}}$ with a linear functional $\mathbb{R}^N \to \mathbb{R}$, we get the following function subspace as hypothesis class for the LS2T

$$\tilde{\mathcal{H}} = \left\{ \langle \sum_{j=1}^N \alpha_j \ell^j, \Phi(\mathbf{x}_1, \ldots, \mathbf{x}_L) \rangle \,|\, \alpha_j \in \mathbb{R} \right\} \subsetneq \mathcal{H} = \left\{ \langle \ell, \Phi(\mathbf{x}_1, \ldots, \mathbf{x}_L) \,|\, \ell \in \mathrm{T}(V) \right\} \tag{14}$$

Hence, we acquire an intuitive explanation of the (hyper)parameters: the width of the LS2T, $N \in \mathbb{N}$ specifies the maximal rank of the low-rank linear functionals of $\Phi$ that the LS2T can represent, while the span of the rank-1 elements, $\mathrm{span}(\ell^1, \ldots, \ell^N)$ determine an $N$-dimensional subspace of the dual space of $\mathrm{T}(V)$ consisting of at most rank-$N$ functionals.

Recall now that without rank restrictions on the linear functionals of Seq2Tens features, Theorem 2.1 would guarantee that any real-valued function $f : \mathrm{Seq}(\mathcal{X}) \to \mathbb{R}$ could be approximated by $f(\mathbf{x}) \approx \langle \ell, \Phi(\mathbf{x}_1, \ldots, \mathbf{x}_L) \rangle$. As pointed out before, the restriction of the hypothesis class to low-rank linear functionals of $\Phi(\mathbf{x}_1, \ldots, \mathbf{x}_L)$ would limit the class of functions of sequences that can be approximated. To ameliorate this, we use LS2T transforms in a sequence-to-sequence fashion that allows us to stack such low-rank functionals, significantly recovering expressiveness.

**Sequence-to-sequence transforms.** We can use LS2T to build sequence-to-sequence transformations in the following way: fix the static map $\phi_\theta : \mathcal{X} \to V$ parametrized by $\theta$ and rank-1 elements such that $\tilde{\theta} = (\theta, \ell^1, \ldots, \ell^N)$ and apply the resulting LS2T map $\tilde{\Phi}_{\tilde{\theta}}$ over expanding windows of $\mathbf{x}$:

$$\mathrm{Seq}(\mathcal{X}) \to \mathrm{Seq}(\mathbb{R}^N), \quad \mathbf{x} \mapsto \left( \tilde{\Phi}_{\tilde{\theta}}(\mathbf{x}_1), \tilde{\Phi}_{\tilde{\theta}}(\mathbf{x}_1, \mathbf{x}_2), \ldots, \tilde{\Phi}_{\tilde{\theta}}(\mathbf{x}_1, \ldots, \mathbf{x}_L) \right). \tag{15}$$

Note that the cost of computing the expanding window sequence-to-sequence transform in equation 15 is no more expensive than computing $\tilde{\Phi}_{\tilde{\theta}}(\mathbf{x}_1, \ldots, \mathbf{x}_L)$ itself due to the recursive nature of our algorithms, for further details see Appendices D.2, D.3, D.4.

**Deep sequence-to-sequence transforms.** Inspired by the empirical successes of deep RNNs (Graves et al., 2013b;a; Sutskever et al., 2014), we iterate the transformation 15 $D$-times:

$$\mathrm{Seq}(\mathcal{X}) \to \mathrm{Seq}(\mathbb{R}^{N_1}) \to \mathrm{Seq}(\mathbb{R}^{N_2}) \to \cdots \to \mathrm{Seq}(\mathbb{R}^{N_D}). \tag{16}$$

Each of these mappings $\mathrm{Seq}(\mathbb{R}^{N_i}) \to \mathrm{Seq}(\mathbb{R}^{N_{i+1}})$ is parametrized by the parameters $\tilde{\theta}_i$ of a static feature map $\phi_{\theta_i}$ and a linear map $\mathcal{L}_i$ specified by $N_i$ rank-1 elements of $\mathrm{T}(V)$; these parameters are collectively denoted by $\tilde{\theta}_i = (\theta_i, \ell_i^1, \ldots, \ell_i^{N_i})$. Evaluating the final sequence in $\mathrm{Seq}(\mathbb{R}^{N_D})$ at the last observation-time $t = L$, we get the deep LS2T map with depth-$D$

$$\tilde{\Phi}_{\tilde{\theta}_1, \ldots, \tilde{\theta}_D} : \mathrm{Seq}(\mathcal{X}) \to \mathbb{R}^{n_D}. \tag{17}$$

Making precise how the stacking of such low-rank sequence-to-sequence transformations approximates general functions requires more tools from algebra, and we provide a rigorous quantitative statement in Appendix C. Here, we just appeal to the analogy made with adding depth in neural networks mentioned earlier and empirically validate this in our experiments in Section 4.

## 4    BUILDING NEURAL NETWORKS WITH LS2T LAYERS

The Seq2Tens map $\Phi$ built from a static feature map $\phi$ is universal if $\phi$ is universal, Theorem 2.1. NNs form a flexible class of universal feature maps with strong empirical success for data in $\mathcal{X} = \mathbb{R}^d$, and thus make a natural choice for $\phi$. Combined with standard deep learning constructions, the framework of Sections 2 and 3 can build modular and expressive layers for sequence learning.

**Neural LS2T layers.**    The simplest choice among many is to use as static feature map $\phi : \mathcal{X} = \mathbb{R}^d \to \mathbb{R}^h$ a feedforward network with depth-$P$, $\phi = \phi_P \circ \cdots \circ \phi_1$ where $\phi_j(\mathbf{x}) = \sigma(\mathbf{W}_j \mathbf{x} + \mathbf{b}_j)$ for $\mathbf{W}_j \in \mathbb{R}^{h \times d}$, $\mathbf{b}_j \in \mathbb{R}^h$. We can then lift this to a map $\varphi : \mathbb{R}^d \to \mathrm{T}(\mathbb{R}^h)$ as prescribed in equation 6. Hence, the resulting LS2T layer $\mathbf{x} \mapsto (\tilde{\Phi}_{\tilde{\theta}}(\mathbf{x}_1, \ldots, \mathbf{x}_i))_{i=1,\ldots,L}$ is a sequence-to-sequence transform $\mathrm{Seq}(\mathbb{R}^d) \to \mathrm{Seq}(\mathbb{R}^h)$ that is parametrized by $\tilde{\theta} = (\mathbf{W}_1, \mathbf{b}_1, \ldots, \mathbf{W}_P, \mathbf{b}_P, \ell_1^1, \ldots, \ell_1^{N_1})$.

**Bidirectional LS2T layers.**    The transformation in equation 15 is completely causal in the sense that each step of the output sequence depends only on past information. For generative models, it can behove us to make the output depend on both past and future information, see Graves et al. (2013a); Baldi et al. (1999); Li & Mandt (2018). Similarly to bidirectional RNNs and LSTMs (Schuster & Paliwal, 1997; Graves & Schmidhuber, 2005), we may achieve this by defining a bidirectional layer,

$$\tilde{\Phi}_{(\tilde{\theta}_1, \tilde{\theta}_2)}^{\mathrm{b}}(\mathbf{x}) : \mathrm{Seq}(\mathbb{R}^d) \to \mathrm{Seq}(\mathbb{R}^{N+N'}), \quad \mathbf{x} \mapsto (\tilde{\Phi}_{\tilde{\theta}_1}(\mathbf{x}_1, \ldots, \mathbf{x}_i), \tilde{\Phi}_{\tilde{\theta}_2}(\mathbf{x}_i, \ldots, \mathbf{x}_L))_{i=1}^L. \quad (18)$$

The sequential nature is kept intact by making the distinction between what classifies as past (the first $N$ coordinates) and future (the last $N'$ coordinates) information. This amounts to having a form of precognition in the model, and has been applied in e.g. dynamics generation (Li & Mandt, 2018), machine translation (Sundermeyer et al., 2014), and speech processing (Graves et al., 2013a).

**Convolutions and LS2T.**    We motivate to replace the time-distributed feedforward layers proposed in the paragraph above by temporal convolutions (CNN) instead. Although theory only requires the preprocessing layer of the LS2T to be a static feature map, we find that it is beneficial to capture some of the sequential information in the preprocessing layer as well, e.g. using CNNs or RNNs. From a mathematical point of view, CNNs are a straightforward extension since they can be interpreted as time-distributed feedforward layers applied to the input sequence augmented with a $p \in \mathbb{N}$ number of its lags for CNN kernel size $p$ (see Appendix D.1 for further discussion).

In the following, we precede our deep LS2T blocks by one or more CNN layers. Intuitively, CNNs and LS2Ts are similar in that both transformations operate on subsequences of their input sequence. The main difference between the two lies in that *CNNs operate on contiguous subsequences*, and therefore, capture local, short-range nonlinear interactions between timesteps; *while LS2Ts (equation 12) use all non-contiguous subsequences*, and hence, learn global, long-range interactions in time. This observation motivates that the inductive biases of the two types of layers (local/global time-interactions) are highly complementary in nature, and we suggest that the improvement in the experiments on the models containing vanilla CNN blocks are due to this complementarity.

## 5    EXPERIMENTS

We demonstrate the modularity and flexibility of the above LS2T and its variants by applying it to (i) multivariate time series classification, (ii) mortality prediction in healthcare, (iii) generative modelling of sequential data. In all cases, we take a strong baseline model (FCN and GP-VAE, as detailed below) and upgrade it with LS2T layers. As Thm. 2.1 requires the Seq2Tens layers to be preceded by at least a static feature map, we expect these layers to perform best as an add-on on top of other models, which however can be quite simple, such as a CNN. The additional computation time is negligible (in fact, for FCN it allows to reduce the number of parameters significantly, while retaining performance), but it can yield substantial improvements. This is remarkable, since the original models are already state-of-the-art on well-established (frequentist and Bayesian) benchmarks.

### 5.1    MULTIVARIATE TIME SERIES CLASSIFICATION

As the first task, we consider multivariate time series classification (TSC) on an archive of benchmark datasets collected by Baydogan (2015). Numerous previous publications report results on this

Table 1: Posterior probabilities given by a Bayesian signed-rank test comparison of the proposed methods against the baselines. $\{>\}$, $\{<\}$, $\{=\}$ refer to the respective events that the row method is better, the column method is better, or that they are equivalent.

| MODEL | LS2T$_{64}^3$ | | | FCN$_{64}$-LS2T$_{64}^3$ | | | FCN$_{128}$-LS2T$_{64}^3$ | | |
|---|---|---|---|---|---|---|---|---|---|
| | $p(>)$ | $p(=)$ | $p(<)$ | $p(>)$ | $p(=)$ | $p(<)$ | $p(>)$ | $p(=)$ | $p(<)$ |
| SMTS (BAYDOGAN & RUNGER, 2015A) | 0.180 | 0.000 | **0.820** | 0.010 | 0.000 | **0.990** | 0.008 | 0.000 | **0.992** |
| LPS (BAYDOGAN & RUNGER, 2015B) | 0.191 | 0.002 | **0.807** | 0.012 | 0.001 | **0.987** | 0.006 | 0.001 | **0.993** |
| MVARF (TUNCEL & BAYDOGAN, 2018) | 0.011 | 0.140 | **0.849** | 0.000 | 0.126 | **0.874** | 0.000 | 0.088 | **0.912** |
| DTW (SAKOE & CHIBA, 1978) | 0.033 | 0.000 | **0.967** | 0.001 | 0.000 | **0.999** | 0.000 | 0.000 | **1.000** |
| ARKERNEL (CUTURI & DOUCET, 2011) | 0.100 | 0.097 | **0.803** | 0.000 | 0.021 | **0.979** | 0.000 | 0.015 | **0.985** |
| GRSF (KARLSSON ET AL., 2016) | 0.481 | 0.011 | **0.508** | 0.028 | 0.013 | **0.960** | 0.022 | 0.013 | **0.965** |
| MUSE (SCHÄFER & LESER, 2017) | 0.405 | 0.128 | **0.467** | 0.001 | 0.074 | **0.925** | 0.001 | 0.077 | **0.922** |
| MLSTMFCN (KARIM ET AL., 2019) | **0.916** | 0.043 | 0.041 | 0.123 | 0.071 | **0.807** | 0.055 | 0.110 | **0.835** |
| FCN$_{128}$ (WANG ET AL., 2017) | **0.998** | 0.002 | 0.000 | 0.363 | 0.186 | **0.451** | 0.169 | 0.011 | **0.820** |
| RESNET (WANG ET AL., 2017) | **0.998** | 0.002 | 0.001 | 0.056 | 0.240 | **0.704** | 0.016 | 0.048 | **0.935** |
| LS2T$_{64}^3$ | - | - | - | 0.000 | 0.001 | **0.999** | 0.000 | 0.001 | **0.999** |
| FCN$_{64}$-LS2T$_{64}^3$ | **0.999** | 0.001 | 0.000 | - | - | - | 0.020 | 0.387 | **0.593** |

archive, which makes it possible to compare against several well-performing competitor methods from the TSC community. These baselines are detailed in Appendix E.1. This archive was also considered in a recent popular survey paper on DL for TSC (Ismail Fawaz et al., 2019), from where we borrow the two best performing models as DL baselines: FCN and ResNet. The FCN is a fully convolutional network which stacks 3 convolutional layers of kernel sizes $(8, 5, 3)$ and filters $(128, 256, 128)$ followed by a global average pooling (GAP) layer, hence employing global parameter sharing. We refer to this model as FCN$_{128}$. The ResNet is a residual network stacking 3 FCN blocks of various widths with skip-connections in between (He et al., 2016) and a final GAP layer.

The FCN is an interesting model to upgrade with LS2T layers, since the LS2T also employs parameter sharing across the sequence length, and as noted previously, convolutions are only able to learn local interactions in time, that in particular makes them ill-suited to picking up on long-range autocorrelations, which is exactly where the LS2T can provide improvements. As our models, we consider three simple architectures: (i) LS2T$_{64}^3$ stacks 3 LS2T layers of order-2 and width-64; (ii) FCN$_{64}$-LS2T$_{64}^3$ precedes the LS2T$_{64}^3$ block by an FCN$_{64}$ block; a downsized version of FCN$_{128}$; (iii) FCN$_{128}$-LS2T$_{64}^3$ uses the full FCN$_{128}$ and follows it by a LS2T$_{64}^3$ block as before. Also, both FCN-LS2T models employ skip-connections from the input to the LS2T block and from the FCN to the classification layer, allowing for the LS2T to directly see the input, and for the FCN to directly affect the final prediction. These hyperparameters were only subject to hand-tuning on a subset of the datasets, and the values we considered were $H, N \in \{32, 64, 128\}$, $M \in \{2, 3, 4\}$ and $D \in \{1, 2, 3\}$, where $H, N \in \mathbb{N}$ is the FCN and LS2T width, resp., while $M \in \mathbb{N}$ is the LS2T order and $D \in \mathbb{N}$ is the LS2T depth. We also employ techniques such as time-embeddings (Liu et al., 2018a), sequence differencing and batch normalization, see Appendix D.1; Appendix E.1 for further details on the experiment and Figure 2 in thereof for a visualization of the architectures.

**Results.** We trained the models, FCN$_{128}$, ResNet, LS2T$_{64}^3$, FCN$_{64}$-LS2T$_{64}^3$, FCN$_{128}$-LS2T$_{64}^3$ on each of the 16 datasets 5 times while results for other methods were borrowed from the cited publications. In Appendix E.1, Figure 3 depicts the box-plot of distributions of accuracies and a CD diagram using the Nemenyi test (Nemenyi, 1963), while Table 7 shows the full list of results. Since mean-ranks based tests raise some paradoxical issues (Benavoli et al., 2016), it is customary to conduct pairwise comparisons using frequentist (Demšar, 2006) or Bayesian (Benavoli et al., 2017) hypothesis tests. We adopted the Bayesian signed-rank test from Benavoli et al. (2014), the posterior probabilities of which are displayed in Table 1, while the Bayesian posteriors are visualized on Figure 4 in App. E.1. The results of the signed-rank test can be summarized as follows: (1) LS2T$_{64}^3$ already outperforms some classic TS classifiers with high probability ($p \geq 0.8$), but it is not competitive with other DL classifiers. This observation is not surprising since even theory requires at least a static feature map to precede the LS2T. (2) FCN$_{64}$-LS2T$_{64}^3$ outperforms almost all models with high probability ($p \geq 0.8$), except for ResNet (which is stil outperformed by $p \geq 0.7$), FCN$_{128}$ and FCN$_{128}$-LS2T$_{64}^3$. When compared with FCN$_{128}$, the test is unable to decide between the two, which upon inspection of the individual results in Table 7 can be explained by that on some datasets the benefit of the added LS2T block is high enough that it outweighs the loss of flexibility incurred by reducing the width of the FCN - arguably these are the datasets where long-range autocorrelations

are present in the input time series, and picking up on these improve the performance - however, on a few datasets the contrary is true. (3) Lastly, $FCN_{128}$-$LS2T^3_{64}$, *outperforms all baseline methods with high probability ($p \geq 0.8$)*, and hence successfully improves on the $FCN_{128}$ via its added ability to learn long-range time-interactions. We remark that *$FCN_{64}$-$LS2T^3_{64}$ has fewer parameters than $FCN_{128}$ by more than 50%*, hence we managed to compress the FCN to a fraction of its original size, while on average still slightly improving its performance, a nontrivial feat by its own accord.

## 5.2 MORTALITY PREDICTION

We consider the PHYSIONET2012 challenge dataset (Goldberger et al., 2000) for mortality prediction, which is a case of medical TSC as the task is to predict in-hospital mortality of patients after their admission to the ICU. This is a difficult ML task due to missingness in the data, low signal-to-noise ratio (SNR), and imbalanced class distributions with a prevalence ratio of around $14\%$. We extend the experiments conducted in Horn et al. (2020), which we also use as very strong baselines. Under the same experimental setting, we train two models: FCN-LS2T as ours and the FCN as another baseline. For both models, we conduct a random search for all hyperparameters with 20 samples from a pre-specified search space, and the setting with best validation performance is used for model evaluation on the test set over 5 independent model trains, exactly the same way as it was done in Horn et al. (2020). We preprocess the data using the same method as in Che et al. (2018, eq. (9)) and additionally handle static features by tiling them along the time axis and adding them as extra coordinates. We additionally introduce in both models a `SpatialDropout1D` layer after all CNN and LS2T layers with the same tunable dropout rate to mitigate the low SNR of the dataset.

**Results.** Table 2 compares the performance of FCN-LS2T with that of FCN and the results from Horn et al. (2020) on 3 metrics: (1) ACCURACY, (2) area under the precision-recall curve (AUPRC), (3) area under the ROC curve (AUROC). We can observe that *FCN-LS2T takes on average first place according to both* ACCURACY *and* AUPRC*, outperforming FCN and all SOTA methods*, e.g. TRANSFORMER (Vaswani et al., 2017), GRU-D

Table 2: Comparison of FCN-LS2T and FCN on PHYSIONET2012 with the results from Horn et al. (2020).

| MODEL | ACCURACY | AUPRC | AUROC |
|---|---|---|---|
| FCN-LS2T | **84.1 ± 1.6** | **53.9 ± 0.5** | 85.6 ± 0.5 |
| FCN | 80.7 ± 1.7 | 52.8 ± 1.3 | 85.6 ± 0.2 |
| GRU-D | 80.0 ± 2.9 | *53.7 ± 0.9* | **86.3 ± 0.3** |
| GRU-SIMPLE | 82.2 ± 0.2 | 42.2 ± 0.6 | 80.8 ± 1.1 |
| IP-NETS | 79.4 ± 0.3 | 51.0 ± 0.6 | *86.0 ± 0.2* |
| PHASED-LSTM | 76.8 ± 5.2 | 38.7 ± 1.5 | 79.0 ± 1.0 |
| TRANSFORMER | *83.7 ± 3.5* | 52.8 ± 2.2 | **86.3 ± 0.8** |
| LATENT-ODE | 76.0 ± 0.1 | 50.7 ± 1.7 | 85.7 ± 0.6 |
| SEFT-ATTN. | 75.3 ± 3.5 | 52.4 ± 1.1 | 85.1 ± 0.4 |

Che et al. (2018), SEFT (Horn et al., 2020), and also being competitive in terms of AUROC. This is very promising, and it suggests that LS2T layers might be particularly well-suited to complex and heterogenous datasets, such as medical time series, since the FCN-LS2T models significantly improved accuracy on ECG as well, another medical dataset in the previous experiment.

## 5.3 GENERATING SEQUENTIAL DATA

Finally, we demonstrate on sequential data imputation for time series and video that LS2Ts do not only provide good representations of sequences in discriminative, but also generative models.

**The GP-VAE model.** In this experiment, we take as base model the recent GP-VAE (Fortuin et al., 2020), that provides state-of-the-art results for probabilistic sequential data imputation. The GP-VAE is essentially based on the HI-VAE (Nazabal et al., 2018) for handling missing data in variational autoencoders (VAEs) (Kingma & Welling, 2013) adapted to the handling of time series data by the use of a Gaussian process (GP) prior (Williams & Rasmussen, 2006) across time in the latent sequence space to capture temporal dynamics. Since the GP-VAE is a highly advanced model, its in-depth description is deferred to Appendix E.3.We extend the experiments conducted in Fortuin et al. (2020), and we make one simple change to the GP-VAE architecture without changing any other hyperparameters or aspects: we introduce a single bidirectional LS2T layer (B-LS2T) into the encoder network that is used in the amortized representation of the means and covariances of the variational posterior. The B-LS2T layer is preceded by a time-embedding and differencing block, and succeeded by channel flattening and layer normalization as depicted in Figure 5. The idea behind this experiment is to see if we can improve the performance of a highly complicated model that is composed of many interacting submodels, by the naive introduction of LS2T layers.

Table 3: Performance comparison of GP-VAE (B-LS2T) with the baseline methods

| METHOD | HMNIST | | | SPRITES | PHYSIONET |
|---|---|---|---|---|---|
| | NLL | MSE | AUROC | MSE | AUROC |
| MEAN IMPUTATION | - | $0.168 \pm 0.000$ | $0.938 \pm 0.000$ | $0.013 \pm 0.000$ | $0.703 \pm 0.000$ |
| FORWARD IMPUTATION | - | $0.177 \pm 0.000$ | $0.935 \pm 0.000$ | $0.028 \pm 0.000$ | $0.710 \pm 0.000$ |
| VAE | $0.599 \pm 0.002$ | $0.232 \pm 0.000$ | $0.922 \pm 0.000$ | $0.028 \pm 0.000$ | $0.677 \pm 0.002$ |
| HI-VAE | $0.372 \pm 0.008$ | $0.134 \pm 0.003$ | $\mathbf{0.962 \pm 0.001}$ | $0.007 \pm 0.000$ | $0.686 \pm 0.010$ |
| GP-VAE | $0.350 \pm 0.007$ | $0.114 \pm 0.002$ | $\mathbf{0.960 \pm 0.002}$ | $\mathbf{0.002 \pm 0.000}$ | $0.730 \pm 0.006$ |
| GP-VAE (B-LS2T) | $\mathbf{0.251 \pm 0.008}$ | $\mathbf{0.092 \pm 0.003}$ | $\mathbf{0.962 \pm 0.001}$ | $\mathbf{0.002 \pm 0.000}$ | $\mathbf{0.743 \pm 0.007}$ |
| BRITS | - | - | - | - | $\mathbf{0.742 \pm 0.008}$ |

**Results.** To make the comparison, we ceteris paribus re-ran all experiments the authors originally included in their paper (Fortuin et al., 2020), which are imputation of Healing MNIST, Sprites, and Physionet 2012. The results are in Table 3, which report the same metrics as used in Fortuin et al. (2020), i.e. negative log-likelihood (NLL, lower is better), mean squared error (MSE, lower is better) on test sets, and downstream classification performance of a linear classifier (AUROC, higher is better). For all other models beside our GP-VAE (B-LS2T), the results were borrowed from Fortuin et al. (2020). We observe that simply adding the B-LS2T layer improved the result in almost all cases, except for Sprites, where the GP-VAE already achieved a very low MSE score. Additionally, when comparing GP-VAE to BRITS on Physionet, the authors argue that although the BRITS achieves a higher AUROC score, the GP-VAE should not be disregarded as it fits a generative model to the data that enjoys the usual Bayesian benefits of predicting distributions instead of point predictions. The results display that by simply adding our layer into the architecture, we managed to elevate the performance of GP-VAE to the same level while retaining these same benefits. We believe the reason for the improvement is a tighter amortization gap in the variational approximation (Cremer et al., 2018) achieved by increasing the expressiveness of the encoder by the LS2T allowing it to pick up on long-range interactions in time. We provide further discussion in Appendix E.3.

## 6 RELATED WORK AND SUMMARY

**Related Work.** The literature on tensor models in ML is vast. Related to our approach we mention pars-pro-toto Tensor Networks (Cichocki et al., 2016), that use classical LR decompositions, such as CP (Carroll & Chang, 1970), Tucker (Tucker, 1966), tensor trains (Oseledets, 2011) and tensor rings (Zhao et al., 2019); further, CNNs have been combined with LR tensor techniques (Cohen et al., 2016; Kossaifi et al., 2017) and extended to RNNs (Khrulkov et al., 2019); Tensor Fusion Networks (Zadeh et al., 2017) and its LR variants (Liu et al., 2018b; Liang et al., 2019; Hou et al., 2019); tensor-based gait recognition (Tao et al., 2007). Our main contribution to this literature is the use of the free algebra $\mathrm{T}(V)$ with its convolution product $\cdot$, instead of $V^{\otimes m}$ with the outer product $\otimes$ that is used in the above papers. While counter-intuitive to work in a larger space $\mathrm{T}(V)$, the additional algebra structure of $(\mathrm{T}(V), \cdot)$ is the main reason for the nice properties of $\Phi$ (*universality, making sequences of arbitrary length comparable, convergence in the continuous time limit*; see Appendix B) which we believe are in turn the main reason for the *strong benchmark performance*. Stacked LR sequence transforms allow to exploit this rich algebraic structure with little computational overhead. Another related literature are path signatures in ML (Lyons, 2014; Chevyrev & Kormilitzin, 2016; Graham, 2013; Bonnier et al., 2019; Toth & Oberhauser, 2020). These arise as special case of Seq2Tens (Appendix B) and our main contribution to this literature is that Seq2Tens resolves a well-known computational bottleneck in this literature since it *never needs to compute and store a signature*, instead it *directly and efficiently learns the functional of the signature*.

**Summary.** We used a classical non-commutative structure to construct a feature map for sequences of arbitrary length. By stacking sequence transforms we turned this into scalable and modular NN layers for sequence data. The main novelty is the use of the free algebra $\mathrm{T}(V)$ constructed from the static feature space $V$. While free algebras are classical in mathematics, their use in ML seems novel and underexplored. We would like to re-emphasize that $(\mathrm{T}(V), \cdot)$ is not a mysterious abstract space: if you know the outer tensor product $\otimes$ then you can easily switch to the tensor convolution product $\cdot$ by taking sums of outer tensor products, as defined in equation 5. As our experiments show, the benefits of this algebraic structure are not just theoretical but can significantly elevate performance of already strong-performing models.

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
