# OpenReview forum: "Seq2Tens: An Efficient Representation of Sequences by Low-Rank Tensor Projections"
_ICLR.cc/2021/Conference — ICLR 2021 Poster_

### Official Review · AnonReviewer3 · 2020-10-20
**An Efficient Representation of Sequences by Low-Rank Tensor Projections**

**Rating:** 5
**Confidence:** 4

**Review:**

The paper presents a method to map static feature space to a space containing sequences of different lengths. The idea is worth of  interest and the appendix gives large amount of information on both theoretical and experimental sides of the work.

The experiments are the main drawback for me of the paper. Indeed, the experiments for sequence discrimination are not convincing enough due to the lack of datasets and framework employed (only the multivariate time series classification problem), and the results obtained. The main results achieved during the TSC classification task are located with moderate or high prior (> 0.6). Moreover, the experiments are conduced with both small models and datasets.

The experiment for sequential data imputation for time series and videos is not clear, and the way of writing and giving explanation is quite confusing for the reader. Since the appendix is large, more details from the appendix in the paper itself will help the reader to more easily follow the idea.

Overall, the experiments have to be larger and considering trickier tasks such as language processing related experiments (speech recognition, language modeling, etc.) that consider sequence mapping to extract robust features.

---

> ### Author Response · Authors · 2020-11-18
> **Our response to Reviewer 3**
>
> Thank you for your comments! Regarding the main criticism,
>
> > (...) experiments have to be larger and considering trickier tasks such as language processing related experiments (speech recognition, language modelling, etc.) (...)
>
> We would like to raise the following points:
>
> _Classification_: the multivariate time series classification archive that we use has become a de-facto standard for time series classification papers. Time series classification is a very large community with many competitive models. For example, the recent and highly cited survey (Fawaz et al 2018, Section 4.1.2) exclusively benchmarks multivariate time series classifiers on this archive. Although the archive is not perfect, it contains datasets from various application domains such as motion, speech, handwriting, etc. as well as time series of various length and dimensions. It makes a challenging and widely used benchmark as witnessed by the cited publications that propose highly specialized architectures to just address this task.
>
> _Generative modelling_: these datasets are very popular in the TS imputation and generative modelling communities. A detailed description of the model and analysis of the experiment is given in Appendix E.2. The GP-VAE is a highly complicated model that builds on advanced concepts such as variational inference, Gaussian processes and several previous works on VAEs, and although we have done our best effort to do so, it is impossible to compress all that work into a few paragraphs. Nevertheless, it is exactly this highly complicated nature of the GP-VAE that the experiment demonstrates the modularity and flexibility of the Seq2Tens map; we were able to improve the performance of a highly complicated model composed of many interacting submodels by the naive plug-in application of our Seq2Tens layer.
>
> We do share the reviewer's enthusiasm about the potential of Seq2Tens in “language modelling, etc”. However, this usually involves very large pipelines with many preprocessing steps and hyperparameters. Applying Seq2Tens in a language model would in our view require a whole paper and partly also address a different target audience. To address the reviewers concerns we have now added a new mortality prediction task, the Physionet 2012 challenge in Section 5.2. This is well-known as a "trickier task" for several reasons 1) missing values in the time series due to irregular and asynchronous sampling, 2) unbalanced class distribution, 3) low signal-to-noise ratio. Again the performance is very strong and competes against recent SOTA methods published in venues like ICML2020, etc.
>
> Finally, a general remark that we should have emphasized more: our primary aim in the experiment section is to demonstrate that using an algebraic structure that is well-known in mathematics but not in ML, the free algebra T(V), yields many benefits; both theoretical and empirical. Further, we show that this can be achieved at little or no additional computational cost, and done in a straightforward and modular way: a vanilla FCN upgraded with Seq2Tens layers beats highly specialized modelsfor time series classification; a popular generative model gets significant performance gains just by adding Seq2Tens layers. That the performance on widely used benchmarks is lifted to SOTA is a nice effect, but should not be the central takeaway of the paper.
>
> > (...) TSC classification task are located with moderate or high prior (> 0.6). Moreover, the experiments are conduced with both small models and datasets
>
> This is not what Table 1 shows: firstly, it shows the posterior (not prior) and, secondly, that our FCN-LS2T^3 beats all of the other models with high-posterior probability, except MLSTMFCN which is slightly beaten by it. Some of the benchmark models are “small”, but not all of them (eg. ResNet and MLSTMFCN). Ultimately the performance matters, and these are the best performing models (see Fawaz et al, 2018). Regarding the size of data sets, we do not consider the video data Sprites, the health data Physionet, and some of the time-series in the classification experiment as small.
>
> Please let us know if this addresses your concerns.

---

### Official Review · AnonReviewer2 · 2020-10-23
**This paper contains some solid contributions but the writing and design need big improvement to communicate and demonstrate the key ideas and benefits.**

**Rating:** 4
**Confidence:** 4

**Review:**

This paper proposes an interesting low-rank tensor representation learning model called Seq2Tens for sequential data. The proposed model can be plugged into existing state-of-the-art neural network models as Seq2Tens layers to improve the performance, which has been demonstrated on several benchmark datasets in this paper.

On one hand, there are some solid contributions made in this paper. On the other hand, I do find it is difficult for the readers to appreciate its merits. Please see my detailed comments below.

1. Possibly due to my limited mathematically background, "the free algebra" in the abstract of the PDF paper is a jargon for me (and possibly other AI/ML readers). Although I could google and learn from Wiki, I am not sure whether it is necessary. In the text abstract online, the authors use "the tensor algebra" in the online abstract of this paper.  Are they interchangeable? Which one is more appropriate?

2. There is no citation to existing works in the first two pages, including the Introduction section. In fact, there is only one citation (on page 3) on the universality in the first three pages of this paper. Does this mean that all other materials are original? Because I am not familiar with so called free algebra, I expect some references to be provided.

3. In the Introduction, after stating the problem of interest, the authors start describing the contributions immediately, without mentioning any existing approaches to solve the problems and giving proper context in judging or appreciating the claimed contributions.

4. The name Seq2Tens is not properly defined, although I can tell it means "sequence to tensors". In fact, in the second paragraph on contributions in Sec. 1 Introduction, the authors did not mention the word "tensor" at all.

5. Section 2 is quite abstract and mathematically, providing some visual illustrations will make the readers' life easier (e.g. cutting example frames in Fig. 3).

6. Equation (3), the degree m is defined to be m>=0. Is it a user-defined hyperparameter? Is there a maximum value for m to take? If yes, what is it and how to determine it in practice? If not, does it mean that m can go to infinity (not likely)?

7. Notations are not standard or consistent. A matrix is denoted as \mathbf(t)_2 and a 3rd-order tensor as \mathbf(t)_3, but the authors also use \mathbf(t) without subscript, in which case I am wondering what is the order (or degree in the authors' chosen terminology). Then in Equation (7), the authors use a non-bold font ell to represent a tensor. In the equation between (1) and (2) (defining Seq2Tens), the right hand side is a capital V without bold, which I assume is a tensor.

8. Acronyms are not always defined before use, e.g. TSC appears in Fig. 1 caption on page 5, but it is only defined on page 6.

9. Section 4 refers too frequent to the appendix and it does not seem to be self-contained. Essential information about the data is missing so readers have to navigate through the 21-page appendix to find information necessary for them to appreciate the methods.

10. Is there any hyperparameter to set? Is there a demonstration of convergence and sensitivity to hyperparameter settings?

11. In Related work, the authors pointed out the vast literature on tensor-based models. However, there is no comparison against any of them. For example, many tensor-based machine learning algorithms have been applied to gait sequences for gait recognition so they model sequences as tensors too (google "sequence tensor gait recognition" w/o quote). These related works should be evaluated against the proposed model, many with public code available.

12. As mentioned above, with many important information put in the appendix, I think the 18 frames displayed in Fig. 3 are more important than those essential information about the data, hyperparameter setting, etc. I am not sure whether human eyes can tell the improvements achieved in much less than 0.001 in MSE on Sprites and whether it matters in practice.

---

> ### Author Response · Authors · 2020-11-19
> **Our response to Reviewer 2 (Part 2)**
>
> > (...) Section 4 refers too frequent to the appendix and it does not seem to be self-contained (...)
>
> We refer to our answer to Reviewer 1, Question 3. This is unavoidable to a certain extent since the paper contains both substantial theoretical and experimental content. We tried to address this as much as possible in the revised version by focusing on the main ideas and addressing the reviewers questions in the main text with precise references to the details in the appendix and the cited papers.
>
> > (...) any hyperparameter to set (...) demonstration of convergence and sensitivity (...)
>
> Yes, there is; we have now added in Section 5 (previously Section 4) the hyperparameter choices that we considered. So far there is no study of sensitivity in the paper, however, we can add such an analysis in the appendix for one of the TSC datasets before the end of the rebuttal period.
>
> > (...) vast literature on tensor-based models (...)
>
>  Indeed there is a vast literature on tensor based models, however, most of these papers are either purely theoretical or their applications are domain specific (e.g. tensor fusion). Ours is a general construction that we demonstrated to work well on a wide range of tasks. We would like to emphasize that we don't explicitly compare against tensor based models, because we benchmark against the best performing models for each given task and dataset. This is a much stronger result.
>
> > (...) sequence tensor gait recognition (...)
>
> We notice that gait recognition is a big area in computer vision and most papers are written with this single task in mind with either huge representation pipelines or highly specialized considerations. Hence, we believe this is outside of the scope of this current work. Nevertheless, we added a citation in the Related work paragraph.
>
> We would like to emphasize that the benchmarks we have chosen are very popular. This can be seen that many of the papers we benchmark against appeared in top conferences and journals.
>
> > (...) I think the 18 frames displayed in Fig. 3 are not more important than those essential information (...)
>
> Thanks, that is a good point. We have now moved the reconstruction frames into the appendix to have more space for expanding on the intuition in the theoretical exposition and including more details from the experiments.
>
> Please let us know if this addresses your concerns.

---

> ### Author Response · Authors · 2020-11-19
> **Our response to Reviewer 2 (Part 1)**
>
> Thank you for your review and comments!
>
> > (...) "the free algebra" (...) "the tensor algebra" (...)
>
> Yes, these are synonyms for the same object T(V). We now consistently refer to T(V) as the free algebra in the revised version. We assume readers will be familiar with tensors and how to multiply tensors with the usual outer product. The definition of T(V) and the tensor convolution product is then simple: it’s just the space of sequences of tensors and the convolution product is given by a simple sum, equation (5). We introduce this at the bottom of p2 and refer to the Appendix for general background on tensors and more details and examples. Please let us know if this clarifies it.
>
> > (...) citation to existing works in the first two pages (...)
>
> We placed the literature review in Section 6 under related work. This is somewhat less common than putting right after the introduction but we thought it fits better there. However, we are also happy to put it in Section 1.
>
> As for novelty, the free algebra with the convolution tensor product is a classical and well-known structure in pure mathematics. Analogously, the use of tensors with the outer tensor product is very popular in ML. In our opinion, the main contribution of this paper, is to show that replacing the outer tensor product with the convolutional tensor product has many advantages, both theoretical and empirical, and that it can be done in an algorithmically efficient way, see Section 6 for a summary.
>
>
> > (...) after stating the problem of interest, the authors start describing the contributions immediately (...)
>
> See our answer above for the theory contribution. For the empirical contribution, we give the precise reference and benchmarks for each experiment in Section 5.
>
> > (...) The name Seq2Tens is not properly defined (...)
>
> Thanks, the term Seq2Tens is now properly defined on page 2 below equation 2. It is the name of the feature map $\Phi$.
>
> > (...) Section 2 is quite abstract and mathematically (...)
>
> We agree and as a response we have considerably revised Section 2 to better explain the key ideas and give intuition by drawing parallels to m-mers subpatterns as used in string kernels. We agree that basic knowledge of tensors is necessary but this is the case for many papers in ML. Nevertheless we give background on tensors with examples in Appendix A. Please let us know if this is more accessible now for someone with less of a math background or where you see more room for improvement.
>
> > (...) the degree m is defined to be m>=0 (...) user-defined hyperparameter (...) maximum value for m to take (...) how to determine it in practice (...)
>
> Yes, this is a hyperparameter. The degree $M \in \mathbb{N}$, as we denote it now in the revised version is called the order of the Seq2Tens layer, and it determines the longest non-contiguous subsequence of the input sequence that the transformation operates on (see eq.8 and 12). The longest subsequence of a sequence $\mathbf{x}=(\mathbf{x}_1, \dots, \mathbf{x}_L)$ is itself, therefore $M \leq L$. We have now added Remark 2.3 that clarifies this.
>  Intuitively, we think of the order as being analogous to the degree of polynomials over vector-valued data, and we experimentally found that it is best kept at a small value ($M \in \{2, 3, 4\}$). Instead of increasing $M$ further, it is most beneficial to increase the depth instead. Also, see our response to Reviewer 4, Question 2 regarding our intuition for the connection between width, order and depth.
>
> > (...) Notations are not standard or consistent (...)
>
> Thanks for pointing these out! We have fixed the subscript issue and now denote an element of $V^{\otimes m}$ by either $\ell_m$ or a subscripted boldfont variable $\mathbf{t_m}$, while the elements of $T(V)$ are either denoted by $\ell$ or a boldfont variable $\mathbf{t}$. We believe the notation $\ell$ is not inconsistent, and that it improves clarity. Many books and papers denote a linear functional by $\ell$, and we use $\ell_m \in V^{\otimes m}$ or $\ell \in T(V)$ to emphasize when it is used as a representer of a linear functional on $V^{\otimes m}$ or $T(V)$.
>
> > (...) the right hand side is a capital V without bold, which I assume is a tensor (...)
>
> The capital V denotes a vector space. First sentence of “Seq2Tens in a nutshell” paragraph: "Given any vector space V we may construct the so-called free algebra T(V) over V."
>
> > (...) Acronyms are not always defined before use (...)
>
> Thanks, we have now fixed this.
>
>
> (Continued next)

---

> ### Comment · AnonReviewer2 · 2020-11-24
> **After reading the response**
>
> I respectively disagree with the authors about the citations. In my opinion, citations should be properly done in sections other than the related works section too.
>
> At least for readers like me, this paper is written in a way that is difficult to appreciate its value.
>
> I did not find it convincing enough to change the rating, though I appreciate the detailed responses from the authors.

---

### Official Review · AnonReviewer4 · 2020-10-28
**Interesting theoretical work!**

**Rating:** 8
**Confidence:** 4

**Review:**

Summary: This paper introduces the free algebra, a classical mathematical concept as a generic tool to represent sequential data of arbitrary length. The proposed method has attractive theoretical property, such as preserving universality of static feature mapping, and convergence in the continuous setting. The author further proposes using stacked rank-1 projection of the free algebra as an approximation to the sequence representation in order to make it computationally feasible neural networks layers. The author illustrated the flexibility and effectiveness of the proposed method by combining the NN implementation with FCN to benchmark on multivariate time series classification problem, and GP-VAE model to benchmark on sequential data imputation problem. The proposed methods shows improved results over previous state-of-the-art.


Significance: This paper provides the community an extension of the universal approximation theorem of NN on sequential data, as well as a generic method to transform static feature maps into sequence features. The experiment shows the proposed method and its implementation is flexible and effective in both discriminative and generative problems. Some questions/feedback: 1. In Proposition 2.3, while the rank-1 projection makes the method computationally feasible, taking sum over all non-contiguous subsequences of x cannot be too cheap? Would the author add analysis on computation complexity here? 2. In the experiment section, as the main motivation of stacking Seq2Tens layers is to mitigate the limitation of the representation power, how does different number of stacked Seq2Tens layers change the model performance?

Clarity: While the paper is highly technical, the author did a good job explaining the idea, concepts and objectives.

Originality: I am not aware of any other work explore free algebra and its usage on sequential data representation.

---

> ### Author Response · Authors · 2020-11-18
> **Our response to Reviewer 4**
>
> Thank you for your review and feedback!
>
> > (...) taking sum over all non-contiguous subsequences of x cannot be too cheap (...)
>
> The formula in question (currently equation 12) seems indeed expensive to compute due to the summation over all sub-sequences. However, it can be cast into a recursive formulation over time that allows us to devise efficient dynamic programming algorithms. This is discussed in Appendices D2, D3 and D4. We should have emphasized this non-trivial fact and we now added a sentence after equation 12 with the resulting big-O bounds.
>
> > (...) number of stacked Seq2Tens layers change the model performance (...)
>
> The Seq2Tens layers are generally robust with respect to the choice of hyperparameters; this is demonstrated by the fact that we were able to use the same architecture with the same hyperparameters on all datasets with minimal initial fine-tuning in the TSC experiment, which had various datasets with the number of examples ranging from a few dozen to several thousand.
> Intuitively, we think of the order parameter M as being analogous to polynomial degree on vector-valued data, i.e. by composing two polynomials of degrees $M_1$ and $M_2$ one gets a polynomial of degree $M_1 \cdot M_2$. Hence, we found that performance was similar between order $M=2$, depth $D=3$ and $M=3$, $D=2$ , but using $M=8$ and $D=1$ performed significantly worse overall. We suggest that this is because depth does not only increase the combined “effective” order, but also compensates for the maximal rank of the low-rank functionals, which is represented by the width parameter $N$ (ending up being completely analogous to the classical width vs depth trade-off in NN). We have an initial statement under Proposition C.3 suggesting the previous, but to make a quantitative statement that explicitly quantifies this trade-off is a challenging problem in algebra.
>
> Please let us know if this addresses your questions.

---

> ### Comment · AnonReviewer4 · 2020-11-23
> **Questions cleared**
>
> Thanks to the authors for the responses to my questions. I continue to hold my earlier rating of clear accept.

---

### Official Review · AnonReviewer1 · 2020-10-29
**The paper uses a classical mathematical tool, i.e., the free algebra to capture this non-commutativity of sequential data, and then utilizes compositions of low-rank tensor projections to reduce the computational complexity.**

**Rating:** 7
**Confidence:** 3

**Review:**

This paper proposes to embed static feature maps into a larger linear space and shows that the proposed method achieves good performance on standard benchmarks. Detailed proofs and theoretical results are given in the appendices. The use of free algebras in ML seems novel and under-explored, although it is classical in mathematics. This paper shows that algebraic structure can significantly elevate the performance of existing models empirically.

Pros:
+This paper shows that the free algebra T(V) can capture the non-commutativity among sequential data theoretically and experimentally, which is under-explored in ML area.

+To address the heavy computational complexity of the free algebra, compositions of low-rank tensor projections(CP decomposition) are employed.


Cons:
-The article is not clearly presented. The introduction is short and without illustrations to show the motivations. There is no intuitive explanation to show how the free algebra improve the models' performance.

-Some notations used in paper are confusing.
a)In Section 3((Stacked) Seq2Tens layers.), what is the difference between D and $D^`$;
b)In Algorithm 1, the truncation degree M and the tensor M share the same notation;
c)In Figure 3, why have the 64$\times$2 S2T and the Seq2Tens blocks different names and shapes? Aren't they the same module？

-A lot of important information, including the details of architecture, algorithms and experiments etc, needs to be obtained from the appendix. This makes the article difficult to understand.

Some typos:
Sec2: a sequences
Appendix D.3, line 5 : be $n_l$ be rank-1 tensors.
Algorithm 1, line8: ·R

---

> ### Author Response · Authors · 2020-11-18
> **Our response to Reviewer 1**
>
> Thank you for your review and comments.
>
> > (...) not clearly presented (...) show motivations (...)
>
> We agree and have now considerably revised Section 2 and Section 3 to address this. We  added more intuition and motivations regarding the first principles based construction of our feature map Seq2Tens. In particular, we draw attention to the new paragraph ‘Some intuition: generalized pattern matching’ on p3.
>
> > (...) explanation to show how the free algebra improve the models' performance (...)
>
> We have now added an additional paragraph in Section 4 (previous Section 3), 'Convolutions and LS2T', which motivates why it is a good idea to combine CNNs with our features, and contains an explanation for the source of improvement on the models containing CNN blocks (which are both FCN and GP-VAE).
>
> > (...) A lot of important information (...) needs to be obtained from the appendix (...)
>
> Yes, but to a certain extent this is unavoidable since the paper contains both substantial theoretical and experimental content. We tried to address this as much as possible in the revised version by focusing on the main ideas and addressing the reviewers questions in the main text with precise references to the details in the appendix and the cited papers.
>
> > (...) Some notations used in paper are confusing (...)
>
> Thank you for pointing these out! We believe that we have resolved in the revised version all mentioned notational clashes and previously ambiguous notation.
>
> > (...) Some typos (...)
>
> Thanks, all corrected.
>
> Please let us know if the above addresses your concerns.

---

### Decision · Program_Chairs · 2021-01-07
**Final Decision**

**Decision:**

Accept (Poster)

**Comment:**

The paper uses free algebras for sequential data representation, and two of the reviewers and the AC find this highly innovative. There were numerous small issues brought up by reviewers (and reviewers disagreed some on the presentation), in particular R3 asking about the experiments, some of which were addressed in the rebuttal.  Overall, because the idea was unusual, it's a bit hard to place and judge this paper. In the end, in the opinion of this AC, the ideas are very creative and there is enough of a chance that this paper could become a very highly cited work, hence we recommend its acceptance.